# Overcoming Depression with 5-HT_2A_ Receptor Ligands

**DOI:** 10.3390/ijms23010010

**Published:** 2021-12-21

**Authors:** Agata Zięba, Piotr Stępnicki, Dariusz Matosiuk, Agnieszka A. Kaczor

**Affiliations:** 1Department of Synthesis and Chemical Technology of Pharmaceutical Substances with Computer Modeling Laboratory, Faculty of Pharmacy, Medical University of Lublin, 4A Chodźki St., PL-20093 Lublin, Poland; piotrstepnicki@umlub.pl (P.S.); dariusz.matosiuk@umlub.pl (D.M.); agnieszkakaczor@umlub.pl (A.A.K.); 2School of Pharmacy, University of Eastern Finland, Yliopistonranta 1, P.O. Box 1627, FI-70211 Kuopio, Finland

**Keywords:** depression, 5-HT_2A_ receptor, antidepressant agents

## Abstract

Depression is a multifactorial disorder that affects millions of people worldwide, and none of the currently available therapeutics can completely cure it. Thus, there is a need for developing novel, potent, and safer agents. Recent medicinal chemistry findings on the structure and function of the serotonin 2A (5-HT_2A_) receptor facilitated design and discovery of novel compounds with antidepressant action. Eligible papers highlighting the importance of 5-HT_2A_ receptors in the pathomechanism of the disorder were identified in the content-screening performed on the popular databases (PubMed, Google Scholar). Articles were critically assessed based on their titles and abstracts. The most accurate papers were chosen to be read and presented in the manuscript. The review summarizes current knowledge on the applicability of 5-HT_2A_ receptor signaling modulators in the treatment of depression. It provides an insight into the structural and physiological features of this receptor. Moreover, it presents an overview of recently conducted virtual screening campaigns aiming to identify novel, potent 5-HT_2A_ receptor ligands and additional data on currently synthesized ligands acting through this protein.

## 1. Introduction

Depression is classified as one of the chronic mental disorders that, according to the World Health Organization data, affects more than 350 million people worldwide [1]. This disorder affects women more often than men and occurs much more frequently in young and elderly people [2]. Many signs may suggest the presence of this condition- the most common somatic symptoms include: persistent sadness or decreased interest in activities considered previously as enjoyable [2], reduced attention, pessimistic view of the future, and overwhelming feeling of guilt or unworthiness. Many patients also experience physical symptoms, such as chronic pain, fatigue, sleep disturbance, or decreased appetite, followed by weight loss and libido decrease [3]. Depending on the severity of the symptoms, patients can be classified as ones suffering from a mild, moderate, or severe form of depression, and, after a diagnosis, they can be advised to start a specific form of treatment [3]. Two main therapeutic strategies include psychological therapy (cognitive behavioral therapy, interpersonal psychotherapy) and pharmacotherapy. Since the probability of the relapse increases with the number of episodes, patients usually undergo prolonged treatment. Unfortunately, no explicit rules determine how long a patient should continue treatment after the first episode of depression [4].

Pharmacotherapy of depression remains an understudied and complex topic since there are many factors involved in the pathomechanism of the disorder. The most widely prescribed medication can be divided into five categories: serotonin reuptake inhibitors (SSRIs), serotonin and norepinephrine reuptake inhibitors (SNRIs), tricyclic antidepressants, monoamine oxidase inhibitors (MAOIs), and atypical antidepressants [5]. The vast majority of the currently applied therapeutics act as direct monoamine concentration modulators or indirect monoamine reuptake inhibitors. Therefore, one can assume that monoamines are one of the most critical substances in the pathomechanism of depression. Medicinal chemistry findings and pharmacological data highlight the therapeutical effectiveness of serotonin concentration modulators in the treatment of mood disorders. Thus, there is a high need to discuss the origins of this neurotransmitter and the importance of the 5-HT_2A_ receptor, since information referring to the rapid antidepressant effect of its ligands have been recently published.

5-hydroxytryptamine is a biogenic amine widely occurring in plants, animals, and the human body that was identified by two independent research groups. An Italian pharmacologist and chemist, Vittorio Erspamer, worked on the histochemical characteristic of enterochromaffin cells, and he discovered that the extract (isolated from examined cells) could contract the intestinal tissue [6]. Almost at the same time, in Cleveland, USA, Maurice M. Rappot, Arda Green, and Irvine Page investigated a vasoconstrictor substance isolated from serum. Finally, it turned out to be the same amine as the one investigated by Erspamer. The substance received the name of serotonin because it was isolated from serum and could affect vascular tone [6,7]. Recently, serotonin has been widely examined due to its numerous pharmacological effects and possible correlation with the pathomechanism of certain diseases and disorders, e.g., depression [8], autism [9], irritable bowel syndrome [10], or schizophrenia [11]. 

In terms of structure, serotonin resembles adrenaline, noradrenaline, and histamine. The bio-synthesis of this neurotransmitter is a two-step process that involves 5-hydroxylation of a tryptophan catalyzed by the tryptophan hydroxylase. This process is followed by the decarboxylation catalyzed by the 5-hydroxytryptophan decarboxylase [12]. The degradation of this bio-amine is dependent mainly on the activity of an enzyme termed monoamine oxidase (MAO). A graphical representation of both processes is depicted in Figure 1.

Serotonin is an amine whose action is mainly linked with the central nervous system (CNS) effects and its neurons are mainly located in the brain stem raphe nuclei [13]. Fibers terminate throughout the CNS, and the transmitter is released by synaptic vesicles. Neuroimaging studies confirm that the highest density of serotonin neurons can be identified in limbic structures—cingulate, entorhinal, insular, and temporopolar regions, along with the ventral and pallidal regions of the striatum and the medial orbitofrontal cortex [14]. High density of serotonin neurons is also identified in the hippocampus. It is worth highlighting that the aforementioned structures resemble a “social brain”, comprising regions important for social cognition and decision-making [15]. Within the CNS, this substance is released into the synaptic cleft and binds to specific proteins to activate 5-HT receptors located on postsynaptic neurons or serotonin autoreceptors located on the presynaptic part of the neuronal membrane. Serotonin and its receptors are believed to regulate, e.g., sexual behavior [15], mood [16], cognition [17], appetite [18], anxiety [8], aggression [19], the release of other neurotransmitters, addiction development [20], and sleep-wake cycles [21].

One of the most valuable pieces of evidence supporting important role of serotonin in the CNS is “tryptophan depletion”, which refers to the condition identified in patients introduced to the low-in-tryptophan (serotonin precursor) diet. Healthy participants did not show symptoms of diminished mood, whereas recovered patients free of medication showed brief, clinically valid symptoms of depression [22]. Further studies suggested that participants who recovered from depression are more likely to suffer from the disorder symptoms after reducing tryptophan intake than those with solid disorder vulnerability [23,24].

It is worth emphasizing that the major concentration of this substance is present peripherally, produced in the gut and stored in platelets. In addition, there is a small amount of free serotonin identified in plasma. Outside the nervous system and gastrointestinal tract, serotonin was identified in organs, such as the lungs [25,26] and kidneys [27,28]. Its influence on physiological processes, such as bowel motility, ejaculatory latency, or urine control, has been widely described [29]. Moreover, an essential role of serotonin was also demonstrated in the development of the hematopoietic system [30] or regulation of bone mass [31].

This multifunctional neurotransmitter exerts its pharmacological effects via numerous receptors. The first serotonin receptors were identified in 1979 by Peroutka and Snyder via radioligand binding methods and were classified into two distinct categories, 5-HT_1_ and 5-HT_2_ [32,33]. Studies conducted by other researchers gave us a better understanding of the structure and classification of these proteins. Current classification distinguishes 14 specialized structures of 5-HT receptors and divides them into seven categories. Apart from the 5-HT_3_ receptor, all belong to the G protein-coupled receptor (GPCRs) superfamily [32] and act through one of Gαi, Gαq/11, or Gαs pathways.

Table 1 gathers information on various 5-HT receptor subtypes and their canonical classification that refers to the affected signaling pathways.

Out of all 14 subtypes, the 5-HT_2A_ receptor has recently gained a lot of interest. This subject has become a hot topic after the recent publication of the Cryo-EM and X-ray structures of the 5-HT_2A_ receptor complexed with agonist, inverse agonist, and partial agonist, which has brought up a discussion on the pharmacological importance of this subtype’s ligands [34,35,36]. This review aims to characterize 5-HT_2A_ receptors and describe their contribution to depression. Additionally, the emphasis is put on the pharmacological significance of 5-HT_2A_ receptor ligands, in depression symptom mitigation. Finally, a brief overview of the recent application of virtual screening campaigns in the development of novel 5-HT ligands is provided.

## 2. Studies on Structure of 5-HT_2A_ Receptor

Before the first X-ray structure of a 5-HT_2A_ receptor was developed, information about the structural features of this protein was obtained from in silico simulations. One of the first papers aimed to determine the 3-D structure of subtype 2A of serotonin receptor was published in 1995 [37]. The authors used data derived from the pharmacophore model of serotonin 2A receptor agonists and antagonists to create a fragmental model depicting the examined structures’ binding properties and affinity data. Moreover, a bacteriorhodopsin structure was used to determine a complete model of the three-dimensional (3D) structure belonging to the 5-HT_2A_ receptor.

In 2000, the crystal structure of a Bovine Rhodopsin (PDB ID: 1F88) was determined using X-ray diffraction, and this is considered a starting point in structural studies on GPCRs [38]. In the following years, researchers took advantage of the availability of this first high-resolution GPCR structure and tried to build a homology model based on this template [39]. More recently, these models were replaced by 3D predictions built on the β2-adrenergic receptor template [40,41,42]. Rapid development in this field provided researchers with more three-dimensional experimentally determined protein structures. That gave the opportunity to choose the best template, characterized by the highest level of identity. A recent publication by Jaiteh et al. aimed to examine homology models of 5-HT_2A_ receptors based on distinct X-ray templates. For this study, authors used 14 structures of β_1A_R, β_2A_R, D_3_R, D_4_R, H_1_R, 5-HT_1B_R, 5-HT_2B_R, 5-HT_2C_R, M_1_R, M_2_R, M_3_R, and M_4_R, Rhodopsin, CXCR4 chemokine A_2_A adenosine, and Cannabinoid 1, characterized by greater and lower identity with a query sequence as molecular modeling templates. Created 3D predictions were submitted into the virtual screening procedure to evaluate whether they can identify known actives among decoys [43]. This procedure contributed to the formulation of new guidelines relevant for GPCR modeling—for templates with >50% identity—, and all should be considered, while those with >30% identity should be evaluated by retrospective virtual screens [43]. While discussing a 3D protein structure determination, it is essential to mention the computational technique that has recently gained a lot of interest. AlphaFold is a deep-learning-based technique that aims to create a three-dimensional representation of a protein without using previously solved protein structures as templates. First, it was introduced in the Critical Assessment of Protein Structure Prediction competition, where it was applied for the T1008 target structure prediction. It has already been shown that, with the AlphaFold approach, it is possible to build reliable and accurate 3D models [44]. Numerous servers have been created that implement the AlphaFold method for the generation of a three-dimensional representation of a protein. “AlphaFold Protein Structure Database” is an example of a commercially available server that gathers protein structures predicted with the use of this method [44].

When it comes to the experimentally determined receptor structures, the first X-ray representations of the 5-HT_2A_ receptor (PDB ID:6A93; 6A94) [45] were determined by Kimura et al. and deposited in Protein Data Bank in 2018. These structures presented an inactive receptor bound to the atypical antipsychotics: risperidone and zotepine, respectively. More recently, another breakthrough in the field of X-ray protein structure determination was made. Two additional X-ray structures of 5-HT_2A_ receptor complexed with hallucinogenic agonist (PDB ID: 6WGT); inverse agonist (PDB ID: 6WH4) and one cryo-EM derived structure of a receptor bound to 25-CN-NBOH (PDB ID: 6WHA) were obtained by Kim et al. [34]. These structures enabled a better understanding of the structure of this GPCR receptor; moreover, they confirmed several in silico derived hypotheses about the structural features of the protein.

The general structure of the 2A subtype of the serotonin receptor resembles other GPCRs and comprises seven transmembrane helices and intracellular amphipathic helix H8. However, two structural features are believed to be extremely important for the proper functioning of a receptor. The first one refers to the bottom hydrophobic cleft located in the ligand-binding pocket. The cleft is surrounded by conserved residues, such as I163^3.40^ and V333^6.45^ in the P-I-F motif, and the W336^4.68^ residue, which is considered an “on-off switch” for the receptor. This receptor also contains a side-extended cavity that connects the orthosteric site and the plasma membrane near the bottom hydrophobic cleft and D155^3.32^, stabilized by a hydrogen bond with Y370^7.43^. D155^3.32^ is another example of a highly conserved residue, present among other GPCRs, essential for the ligand binding [44]. Mutations located in the neighborhood of the D155^3.32^ residue lead to the loss of function for almost all 5-HT_2A_ ligands. On the other hand, S159^3.36^ seems to be important for drug binding, as well, since it is involved in the anchoring of the charged terminal amine moiety of 5-HT and other ligands [41]. The previously mentioned side-extended cavity is surrounded by conserved residues located on TM3, TM4, TM5, and the extracellular loop 2. The G238^5.42^ residue located at the entrance is considered to be essential for the formation of the cavity. Moreover, this glycine residue is considered a feature characteristic only for the 5-HT receptors [45].

5-HT_2A_ receptor activation models suggest that it is capable of creating a wide range of ligand-dependent structural responses [46]. This property is also known as “functional selectivity” and is a common feature among many GPCRs. Functional selectivity, understood as protein-mediated activation of G_αq/11_ or pertussis toxin-sensitive G_i/o_ protein, has been considered in terms of this protein’s interactions with hallucinogenic and non-hallucinogenic agonists. Moreover, the functional selectivity of the 5-HT_2A_ receptor also refers to its connections with the β-arrestin-dependent signaling pathways [47]. Many studies have been performed in order to provide detailed information about this interesting phenomenon, and it turned out that differences in the conformation of binding pocket residues can be identified depending on the type of examined ligand. In silico studies performed by Perez-Aguilar et al. revealed that the second intracellular loop (ICL2) plays a key role in the receptors’ interaction with G-protein. Thus, β-arrestins and distinct ligands affect this loop differently. When 5-HT_2A_R is bound to the hallucinogen (e.g., LSD), ICL2 loop prefers more outward-upward conformations. On the other hand, when this receptor is complexed to the non-hallucinogen drug or present in an unbound form, the unique conformation of a second loop is not that frequently observed. Such conformation may be regulated by the extent of the interaction between D172^3.49^ (from the conserved DRY motif, TM3) and H183^3.52^ (from ICL2) [47]. Recent experimental studies conducted by Kim et al. confirmed that, in the LSD-activated structure of a 5-HT_2A_ receptor, a second extracellular loop forms a lid-like shape that prolongs ligands residence time [34].

Similarly, as in other activated GPCRs, agonist binding leads to a contraction of the extracellular binding pocket and expansion of the intracellular end. That creates additional space for transducers, such as G-proteins or arrestins [34]. Other fragments necessary for the activation have been identified, as well, and are believed to be located in the receptors conserved motifs. Those are an inward shift of residues from the NPxxY motif, rearrangement of R173^3.50^ residue from the E/DRY motif that breaks the ionic lock formed between R173^3.50^ and E318^6.30^. Additionally, modification in the P-I-F motif, involving rotation of the side chain of W336^6.48^ and subsequent movement of the F332^6.44^ side chain, is considered essential for the receptor activation and signal transduction [29]. Kim et al. also cautiously examined the binding poses of another 5-HT_2A_ receptor ligand [34]. Interestingly, a 25-CN NBOH, a selective receptor agonist, showed a unique pose in a receptors binding pocket. The 2-hydroxyphenyl moiety of a ligand entered a pocket formed between TM3/TM6 and interacted with the indole ring of W336^6.48^. That interfered with a large displacement of the W336’s side chain and acted as a pivot for the outward movement of TM6. Moreover, an edge-to-face π–π interaction of a ligand with the previously mentioned residue, hydrogen bond with S159^3.36^, accommodation by the conserved G369^7.42^ are unique for 25CN-NBOH binding. Furthermore, these features can be considered a possible reason for its agonistic selectivity toward the 2A subtype of serotonin receptor [34].

On the other hand, examining the 5-HT_2A_ receptor complexes with commonly used antipsychotics, zotepine and risperidone, enabled a better understanding of the structural changes that occur during the antagonist binding. Both antipsychotics create a salt bridge between D155^3.32^ and a basic nitrogen atom of their molecules. Moreover, their fluorobenzisoxazole ring and benzene ring are located in the bottom hydrophobic cleft, and they interact via CH- π with S159^3.36^. Additional hydrophobic interactions are formed between I163^3.40^, F243^5.47^, F332^6.44^, and those rings. Some other edge-to-edge interactions with W336^6.48^ and F340^6.52^ have been identified as necessary for the ligand binding. Close contacts between ligands and fragments of the P-I-F motif are believed to block the rearrangements of mentioned residues and stabilize the structure in an inactive state.

## 3. 5-HT_2A_ Receptor Is It the Right Target?

The serotonin 5-HT_2A_ receptor belongs to the A-class of G-protein coupled receptors, which are considered one of the most important therapeutic targets. Human subtype 2A of the 5-HT receptor was cloned in the 1990s and is characterized by 41% and 46% of identity with the 5-HT_2B_ and 5-HT_2C_ subtypes. It is worth highlighting that it shares more than 90% of sequence homology with the rat 5-HT_2A_ receptor, but some significant differences in their functional selectivity, regulation of desensitization, and resensitization can be identified [35]. In humans, this receptor is encoded by the HTR2A gene, located on chromosome 13 (46.83–46.9 Mb) [35]. Out of all 14 subtypes of serotonin receptors, the 5-HT_2A_ is widely abundant across the central nervous system, with the highest concentration in monoaminergic brainstem levels, such as the median raphe nucleus/dorsal raphe nucleus, the locus coeruleus, and ventral tegmental area—structures involved in the regulation of mood. Moreover, this receptor acts as a cognitive process modulator by indirect or direct influence on glutamate release and interaction with other targets, such as 5-HT_1A_, GABA_A_, adenosine A_1_, and OX_2_ receptors [48,49,50,51]. More recent findings suggest that specific polymorphisms in nucleotides encoding 5-HT_2A_ receptor structure may act as a potential biomarker for antidepressant early response [24]. That confirms the strong connotation between alterations in serotonin concentration and the occurrence of the disorder.

As it was previously mentioned, the 5-HT_2A_R has been shown to interact with β-arrestin-1 and β-arrestin-2, both in vitro and in vivo [52]. A study conducted by Gelber et al. has shown that β-arrestin-2 knock-out mice, in which 5-HT_2A_ receptors predominantly face the cell surface, were incapable of showing behavioral responses [53]. It would suggest that β-arrestin-2 is involved in the mediation of intracellular trafficking of the 5-HT_2A_ receptors, and cellular events may be necessary in the induction of specific behavioral responses to elevated serotonin concentration [54]. However, in β-arrestin-2-knock-out mice treated with DOI, a preferential 5-HT_2A_ receptor agonist, a desired behavioral effect of head-twitch was obtained [54,55,56]. That suggests β-arrestins are not essential for the induction of DOI-mediated response. Thus, identification of the nature of the ligand seems to be crucial for the determination of the receptor signaling pathway and predicting the physiological response. Nonetheless, it became clear that serotonin 2A receptors coupling to the β-arrestins can lower or elevate GPCR signaling, which means that GPCR signaling may differ depending on the type of ligand.

5-HT_2A_ receptors can regulate their signaling by forming stable hetero- and homodimeric complexes with other GPCRs (e.g., mGluR2, D2-DA Rs). Although the in vivo implications of this property have not been yet determined, this process is considered to affect the binding and coupling properties of the receptor. This intriguing property has been supported by the disappearance of the head-twitch response in mGlu2-knock-out animal models treated with LSD, a preferential 5-HT_2A_ receptor agonist [54,57,58]. Moreover, allosteric receptor-receptor interactions in a 5-HT_2A_ and 5-HT_1A_ receptor complex play a vital role in mood modulation. Activation of the 5-HT_2A_ protomer seems to result in a postjunctional 5-HT_1A_ protomer signaling decrease in the forebrain. Such action is believed to favor the development of depression. Therefore, it is essential to examine these complexes in terms of their impact on disorder modulation [30]. Since this complex was reported as the one capable of disappearing from the pyramidal cell layer (CA-1, CA-2 area) in animals examined 24 h after participation in the forced swim test, it is considered as a dynamic one. A possible explanation of this phenomenon is related to the negative effect of stress, increased glucocorticoid release, alteration in gene transcription, and modification in protein composition, resulting in changes in the density of homo- and heteroreceptor complexes [59].

It is worth highlighting that 5-HT_2A_Rs can form heterocomplexes with oxytocin receptors. Both receptors share a similar distribution pattern and have been identified primarily in cortical and subcortical areas of the forebrain. Oxytocin can create an anxiolytic response through its receptors (OXTRs) located in serotonin neurons [30]. In this heterocomplex, the 5-HT_2A_ protomer possesses a dominating role and decreases the OXTR-mediated G_αq_ signaling. This interaction may also be involved in the disorder’s pathomechanism since attenuation of oxytocin transmission results in mood and social behavior decrease. There is also evidence that once this mechanism is put into action, which means agonists activate the 5-HT_2A_ protomer, the oxytocin transmission seems to be no longer dependent on the activation of the serotonin receptor component. It seems as if a constitutive activity has been developed, which maintains allosteric inhibition of the oxytocin component [30,60,61]. Additionally, when an antagonist is bound to a 5-HT_2A_ protomer, the OXTR signaling could not be restored.

In this divagation on whether or not 5-HT_2A_ is an important molecular target in the treatment of depression, one should remember that the several decades-long research on pathomechanism and pharmacotherapy of this condition revealed that some receptor subtypes should be blocked (e.g., 5-HT_2A,_ 5-HT_2C_). In contrast, the others should be activated (e.g., 5-HT_1A,_ 5-HT_1B_) to obtain the most satisfying therapeutic results. In the case of the serotonin receptor 2A, both agonists and antagonists are believed to possess desired properties potentially valuable in the treatment of mood disorders.

Antagonists of the 5-HT_2A_ receptor represent a significant category of medication, also widely used in the pharmacotherapy of mental diseases. This group is represented by some of the commonly used antipsychotics, such as clozapine, olanzapine, and risperidone, that are pretty efficient in reducing both positive and negative symptoms of schizophrenia [62]. Regarding their antidepressant properties, one explanation of their efficacy is that 5-HT_1A_ and 5-HT_2A_ receptors share a similar distribution pattern. Thus, the generation of nerve impulses is regulated in the opposite manner by 5-HT_2A_ and 5-HT_1A_ receptors. Therefore, a blockade of the 2A subtype of serotonin receptor may enhance the biological activity of the second subtype, which will lead to the opening of the G-protein-coupled inwardly-rectifying potassium channels. Such action will result in inhibition of neuronal firing and hyperpolarization, which is extremely important since limbic networks are hyperactive in major depressive disorder [30]. It is worth highlighting that numerous preclinical and clinical studies confirmed the usefulness of 5-HT_2A_ receptor antagonists in the treatment of depression; Pandey et al. evaluated the effect of 5-HT_2A_ receptor antagonist-BIP-1 administration. In both acute and sustained administration, this substance caused an antidepressant effect in animal models of depression [63]. Such findings confirmed the effectiveness of these compounds in depression symptom mitigation. Moreover, when serotonin 2A receptor antagonists are co-administered with SSRIs, their antidepressant activity in patients suffering from major depression increases significantly [64,65]. Examination of selective 5-HT_2A_ receptor antagonist–M100907, revealed that it also elevates the pharmacological response to SSRIs in the study termed the differential reinforcement of low rate 72 s schedule. However, improvement in executive functions is not a result of the SSRI-induced 5-HT concentration elevation, but it is correlated with the blockade of the 5-HT_2A_ receptor itself [66]. A similar, rapid antidepressant effect was also observed in treatment-resistant patients medicated with monoamine oxidase inhibitors when combined with 5-HT_2A_ receptor antagonists [67].

On the other hand, a group of 5-HT_2A_ receptor agonists has been reported to be helpful in the treatment of various types of depression. This category of ligands comprises, e.g., psilocybin, 25-NBOH, and mexamine. Recent studies emphasize the importance of 5-HT_2A_ receptor agonists in the therapy of treatment-resistant forms of the disorder. The exact mechanism of their action remains unknown. However, according to more recent fMRI studies, psychedelics-psilocybin, in particular, decrease the blood flow in certain brain regions partially overlapping default mode network and fronto-parieto-termporal network correlated with spontaneous behavior and introspection. Psilocybin studies also revealed that this compound enhances signaling and disrupts the functional connectivity within limbic system neurons. Recent studies suggest that psychedelics promote functional and structural plasticity in prefrontal cortical neurons. Common hallucinogens promote the growth of neurons and dendric spines, leading to the so-called “brain reset” and the rapid antidepressant effect [68]. Projects aiming to examine the effects of psychedelics on mood, social cognition, and anxiety revealed that acute LSD administration reduces the activity of the left amygdala and the right part of the prefrontal cortex in healthy subjects exposed to the presentation of fearful faces. On the other hand, psilocybin regulates the threat-induced signaling from the amygdala to the primary visual cortex. In other studies, the same substance was found to decrease the connectivity between the amygdala and striatum in patients submitted into the happy face discrimination test. However, the study aiming to identify depressed-patients amygdala response to fearful, happy, or neutral stimuli showed an increased response to the fearful and happy face after administering a single dose of psilocybin. This suggests that the discussed substance does not disrupt the emotional response, but it can help patients confront their fears in the controlled environment. Other studies investigating the therapeutic value of psilocybin and other psychedelics suggest that they reduce the feelings associated with negative social interactions, such as rejection; increase the personal meaning to previously meaningless objects; and activate the visual regions of a brain comparable to the images experienced with opened eyes. All these findings bring hope that, when combined with professional psychological treatment, psychedelics can help cure thousands of patients suffering from depression, phobia, trauma, etc. [69].

In the last decade, the quantity of papers focusing on this field is growing; a study published by Hesselgrave et al. describes a trial conducted on chronically stressed animal models, which were medicated with a single injection of psilocybin. The administered dose reversed anhedonic responses and restored synaptic strength of excitatory neurons located in cortico-mesolimbic brain regions. In addition, neither electrophysiological nor behavioral responses to psilocybin injection were canceled by pretreatment with ketanserin (the 5-HT_2A/2C_ antagonist) [70]. Furthermore, a study released by Raval et al. evaluated the effect of psilocybin on the structure of a pig’s brain. Animals were divided into two groups, out of which only one received the agonist-containing injection. Half of the representatives from both groups were euthanized on the first day post-treatment, whereas the second half was euthanized a week post-injection. Post-mortem examination of animals’ brains revealed that, in the individuals euthanized one day post-psilocybin injection, the hippocampal density of synaptic vesicle protein increased by 4.42%, and the density of 5-HT_2A_ receptors located in the hippocampus and prefrontal cortex decreased. In the case of animals euthanized 7-days post injections, in their brains, there was a significant increase in SV2A density in both prefrontal cortex and hippocampus, whereas no differences in density of 5-HT_2A_ receptors were discovered. These results suggest that psilocybin causes persistent synaptogenesis and a noticeable decline in 5-HT_2A_ receptor density, which may be correlated with its antidepressant properties [71].

A systematic review published by Romeo et al. gathered information from various clinical trials performed between 1990–2020, which confirmed the effectiveness of psychedelics in the treatments of drug-resistant depression, as well as in other psychiatric and addictive disorders. Moreover, presented data indicated that the intensity of the acute psychedelic administration could be correlated with the effectiveness of symptom mitigation in patients suffering from treatment-resistant depression [72]. In this review, authors also determined some factors that could negatively affect the effectiveness of pharmacotherapy; it turned out that the number of unsuccessful therapies patients went through contributed to a poorer prognosis of the potential recovery. The authors highlighted the importance of non-pharmacological factors (e.g., environment in which the therapy was conducted or persons’ expectations) in the efficacy of psychedelics treatment. Moreover, they suggested several potential mechanisms through which psychedelics could act their pharmacological effects [73]. Another meta-analysis published by Romeo et al. presented evidence on kinetics-related aspects of psychedelic treatment in the mitigation of depressive symptoms. Eight studies with a significant decrease in depressive symptoms were evaluated. In these trials, a temporary heart rate, blood systolic, and diastolic pressure were observed; however, no severe adverse reactions were reported. This confirms the safety of 5-HT_2A_ receptor agonists in the treatment of depression [73].

## 4. Novel 5-HT_2A_ Ligands as Antidepressant Agents–Agonists Emerge from the Shadows of Antagonists?

Searching the medicinal chemistry literature for the reports of novel ligands of serotonin 5-HT_2A_ receptor with antidepressant-like properties, it will be readily seen that the interest is majorly focused on antagonists of this target. There is a considerable number of scientific papers on newly designed 5-HT_2A_ antagonists with antidepressant potential in the more recent five years. Here are given a few examples of such reports. Kim et al. proposed compounds based on phthalazinone scaffold, acting as 5-HT_2A_ and 5-HT_2C_ antagonists, with an affinity for serotonin transporter [74]. Another research group designed a novel arylpropylamine derivative as an inhibitor of serotonin and noradrenaline reuptake with an antagonist activity toward the 5-HT_2A_ receptor [75]. Evans et al. synthesized thioadatanserin, an analog of adatanserin, and its dialkylated derivatives, antagonists of 5-HT_2A_, and partial agonists of 5-HT_1A_ receptors [76].

In this part, we emphasize the development of 5-HT_2A_ agonists, as an insufficiently investigated, yet promising agents for the development of new therapies of depression. Agonists of the 5-HT_2A_ receptor are suggested to be capable of producing a long-lasting therapeutic effect in depressive disorders by leading to the increase in the neuronal growth in the anterior structures of the brain, such as the prefrontal cortex [68]. Affecting the plasticity of the neural circuits reverses the pathological changes within these structures, thus exerting a more sustained therapeutic effect than just reducing the symptoms of the disorder [77]. However, due to the fact that activating the 5-HT_2A_ receptor is frequently associated with the occurrence of hallucinogenic effects, this field in drug discovery is still underexplored. The reports of novel promising drug candidates against depression in the group of 5-HT_2A_ agonists are very limited. In fact, as a result of searching for articles from the most recent five years on novel agonists of the being discussed receptor in the PubMed database (using Mesh Terms: ‘5 HT_2A_ agonist’ and ‘agents, antidepressive’ or ‘depression’), only one record is returned. The publication by Cameron et al. reveals novel non-hallucinogenic compounds, acting as a potent 5-HT_2A_ agonists, which are analogs of a psychedelic, ibogaine (Figure 2), derived from the plant *Tabernanthe iboga*. In order to elucidate which part of the ibogaine chemical structure is responsible for promoting neural growth, the research group performed function-oriented synthesis. Subsequent deletion of one of the key structural feature of the compound allowed for definition of the pharmacophore model for psychoplastogenic properties of ibogaine. It turned out that the analogs without tetrahydroazepine moiety did not stimulate neuronal growth, but those with removed isoquinuclidine and retained tetrahydroazepine ring remained active. One of the derivatives, ibogainalog (IBG) (Figure 2), exhibited comparable psychoplastogenic activity to ibogaine; thus, it was chosen for further optimization [78].

In the next step, the authors attempted to design ibogaine analog devoid of hallucinogenic properties. They relied on the fact that the agonist of 5-HT_2A_ receptor, 5-methoxy-*N*,*N*-dimethyltryptamine (5-MeO-DMT), is a potent hallucinogen, but its analog with methoxy substituent shifted from the position five to six does not show such activity. Transferring this structural modification to ibogainalog (IBG) resulted in obtaining the derivative tabernanthalog (TBG) (Figure 2), named based on similarity to another alkaloid found in *Tabernanthe iboga*–tabernanthine (Figure 2), with the 5-methoxyindole moiety replaced with 6-methoxyindole fragment. In order to evaluate the potential hallucinogenic activity of IBG and TBG, the head-twitch response test, a popular test used in the assessment of the activation of 5-HT_2A_ receptor, was performed. The results indicated that IBG displays reduced hallucinogenic properties, while TBG shows a lack of this effect when compared to the 5-MeO-DMT used as a positive control. These findings confirmed the hypothesis that the slight modification of the methoxy substituent position may result in obtaining non-hallucinogenic analogs of ibogaine. Another advantage of the derivatives deprived of isoquinuclidine fragment over ibogaine is their reduced lipophilicity and, therefore, lower risk of inducing toxic effects, particularly cardiotoxicity. Ibogaine has high tendency to accumulate in the adipose tissue due to its high lipophilicity and, hence, contributes to the toxic effects on the cardiac system through the inhibition of hERG potassium channels [79,80]. Treatment with both IBG and TBG does not lead to arrhythmias and causes significantly fewer malformations and deaths in the zebrafish model comparing to ibogaine [78]. Not without significance is the simplicity of the synthesis of IBG and TBG, which both can be produced in one-step synthesis on a large scale. In contrast, known routes of ibogaine synthesis consist of several steps and result in very low overall yields [81].

Subsequently, Cameron et al. assessed the effect of TBG (as it displayed a better safety profile than IBG) on dendritic growth. They proved that the treatment with TBG increases dendritic arborization of the rat cortical neurons. This effect is suggested to be associated with the activation of 5-HT_2A_ receptor since the pretreatment with ketanserin, a 5-HT_2A_ receptor antagonist, suppresses this response. Additionally, TBG leads to an increase in density of dendritic spines, with the dynamics of spine formation comparable with 2,5-dimethoxy-4-iodoamphetamine (DOI), the hallucinogenic agonist of the 5-HT_2A_ receptor.

As already mentioned, the antidepressant effect of 5-HT_2A_ agonists is believed to emerge from the improvement of neural plasticity mediated by the activation of this receptor. In order to evaluate TBG as a potential antidepressant and to compare its activity with antidepressant, ketamine, a forced swim test was performed. On the first day of the experiment, a pre-test had been performed, and, 24 h after, the tested compounds were administered. Then, the forced swim test was performed 24 h and seven days after injection of the substances. Both TBG and ketamine reduced the time of immobility to a significant extent 24 h after administration, while, after one week, the effect of ketamine seemed more sustained compared with TBG. As expected, the antidepressant effect of TBG was blocked by the treatment with ketanserin, a 5-HT_2A_ antagonist, which confirms the role of 5-HT_2A_ receptor activation in producing the antidepressant-like responses. The effect of TBG on other behaviors related to depressive disorders should be evaluated in further studies, as pointed out by the authors [78]. Nevertheless, it may be concluded that these findings will serve as an incentive to give more consideration to the activation 5-HT_2A_ receptor as a new strategy to combat depression.

## 5. In Silico Methods Aiming to Identify Novel 5-HT_2A_ Receptor Ligands

A wide variety of physiological connotations made the 2A subtype of serotonin receptor an important biological target, especially useful in the treatment of mental diseases. Recently, in silico methods are inevitable in the process of drug development. These techniques allow examination and integration of data from various sources. One of the most significant advantages of in silico drug discovery is reducing the overall costs and efforts related to extensive research that have to be carried before a specific medication can be introduced to the market. Therefore, these techniques are widely applied in the studies aiming to develop molecules characterized by a high affinity to a specific molecular target, with minimal side effects and promising pharmacokinetic parameters (ADME). Therefore, a combination of ligand- and structure-based approaches can be implemented to obtain reliable predictions useful in the identification of novel drug-like molecules [82]. In this part, we would like to focus on virtual screening approach, that sheds a light of hope on the rapid development of new, potentially valuable for the treatment of depression medication.

Virtual screening campaigns belong to the computational methods, which automatically explore large libraries of drug-like molecules to identify compounds most likely to possess a desired pharmacological profile. This strategy can be visualized as a funnel that aims to select the most promising molecules to be submitted into in vitro assays [83]. Depending on the type of available information, two major approaches can be distinguished in this context [83]. The first one-ligand-based virtual screening takes advantage of a dataset collecting compounds with confirmed activity toward a desired molecular target. This approach is applied when the structure of the molecular target is unknown or desired protein was resolved in the apo form [84]. Usually, this technique utilizes a pharmacophore model based on biologically evaluated active compounds that provide indirect information about interactions of ligands with the molecular targets binding site. The fundament of shape-based similarity screening refers to the rule stating that molecules possessing similar chemistry and structure to the active templates carry a significant probability of sharing a desired pharmacological profile. Therefore, ligands differing from the query sequence are more likely to be rejected during the virtual screening procedure [85,86]. Many research groups have already implemented this strategy, including Zhang et al., who examined a group of 35 derivatives possessing antagonistic activity toward D_2_ and 5-HT_2A_ receptors. They used these compounds to create a QSAR model to evaluate the structure-activity relationships of these molecules. Moreover, researchers constructed a pharmacophore model depicting features most important for ligand binding. Later, this pharmacophore model was utilized in the screening study, performed on the ZINC12 database with UNITY Flex software. After final refinement and filtering, four compounds were selected for in silico molecular optimization and basic pharmacokinetics properties prediction [87]. An interesting approach was also used by Kumar et al., who utilized 2D similarity, 3D dissimilarity, and a fusion of 2D/3D similarity data and filtered the ZINC database. Retrieved complexes were docked to the structure of the 5-HT_2A_ receptor and evaluated with the use of molecular dynamics (MD) methods. This experiment resulted in the identification of several residues important for the ligand binding and 287 potential 5-HT_2A_ receptor antagonists that may be submitted into additional experimental validation [88].

The latter technique, known as structure-based virtual screening (SBVS), takes advantage of a currently available 3D structure of a 5-HT_2A_ receptor or any other biological target. It aims to find the best interaction mode between the target and the compound, and it applies different scoring functions to estimate the possibility of a stable protein-ligand complex formation [83]. In this procedure, screened ligands are sorted according to the calculated affinity to the receptor’s binding site, allowing us to identify the compounds with the most promising pharmacological activity [89]. The biggest drawback of this approach is usually related to choosing the best scoring function in order to identify a ligand in the target’s binding site, as well as the development of an appropriate scoring function to estimate the strength of the non-covalent interaction occurring between the analyzed compounds and molecular target [83].

A work recently published by Wang et al. outlined results obtained from detailed inspection of a 5-HT_2A_ receptor and its structure application in the identification of the novel 5-HT_2A_ receptor antagonists. The structure-based pharmacophore model was developed in LigandScout 4.2 and contained features essential for the proper binding of the ligand, such as information on the desired localization of the hydrogen bond acceptors, aromatic rings, etc. The retrieved model was used in the Schrödinger’s suite for the virtual screening of TCMNP and FDA databases. Returned hits were evaluated and compared based on their docking scores, binding modes, free energies, and their MD simulation performances–that led to the identification of three potential HITS. The toxicity and basic pharmacokinetic properties of these compounds were also evaluated [90]. Staroń et al. published a paper describing a study attempting to identify novel 5-HT_6_/5-HT_2A_ ligands. In this case, researchers followed the “one-drug–multiple targets” methodology and used 2D fingerprint screening to obtain 5-HT_6_R ligands. Later, these compounds were evaluated using in silico methodology to determine basic ADMET properties. The remaining compounds were submitted into 3D pharmacophore screening and flexible docking study to consequently narrow the set of examined hits. Chosen molecules were evaluated with experimental techniques in order to examine their affinity toward desired molecular targets [91].

Gandhimathi et al. also performed structure-based virtual screening to identify novel ligands acting as 2A subtype receptor agonists or antagonists. For this purpose, he took advantage of the X-ray structure of a β_2_-adrenergic receptor (PDB ID: 3SN6) to create two homology models representing both the active and inactive state of the protein. Created models were used for the virtual screening, performed using a Zinc database. Additional criteria were implemented to narrow the retrieved compound set, such as compatibility with Lipinski’s rule of five. Hits were introduced to the induced-fit docking procedure, and obtained docking poses were compared with well-established ones. This study identified 15 molecules that were believed to possess desired 5-HT_2A_ agonistic or antagonistic features [41].

Thus, it can be observed that virtual screening has already been successfully implemented in the development of potentially valuable ligands.

## 6. Conclusions

In this review, we summarized fundamental knowledge on the structure and functions of the 5-HT_2A_ receptor. Moreover, we have discussed its numerous connotations with the pathomechanism of depression and enumerated a plethora of reasons confirming its potential usefulness in the treatment of this disorder. Analysis of the collected data ensured that both agonists and antagonists of the 5-HT_2A_ receptor are important in the treatment of mood disorders.

This review also highlighted the role of computational techniques in the development of novel therapeutic agents. Out of all techniques, virtual screening campaigns seem to provide promising results in the identification of novel medication acting through the desired target or possessing a desired structural pattern. However, there is still an observable discrepancy between the quantity of experimental data corresponding to antagonists and agonists of this receptor. Thus, we believe that information presented in this work will be an incentive to look for novel molecules possessing agonistic activity toward the 2A serotonin receptor.

With all this information taken into account, the 5-HT_2A_ receptor seems to be an attractive molecular target, and novel ligands acting through this protein may be helpful in the treatment of depressive disorders.

## Figures and Tables

**Figure 1 ijms-23-00010-f001:**
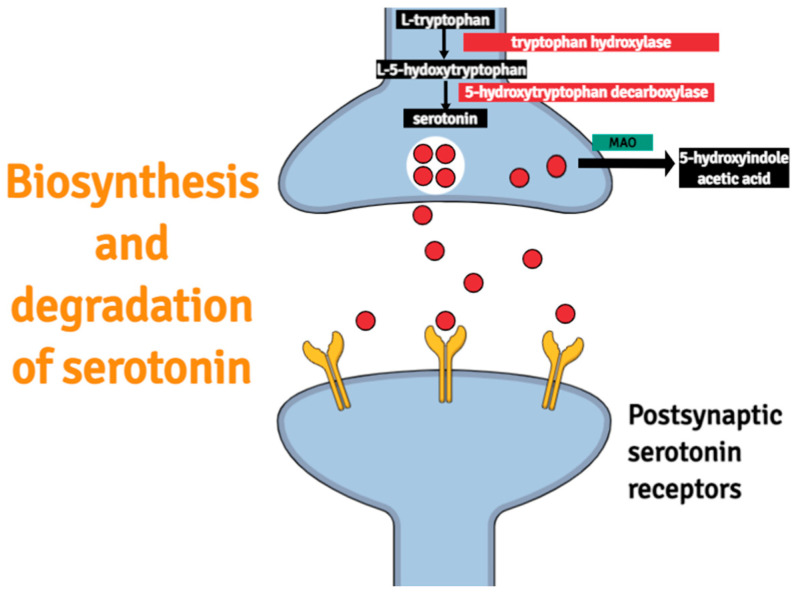
A scheme presenting a graphical interpretation of biosynthesis and degradation of serotonin. MAO stands for monoamine oxidase, an enzyme mainly responsible for the degradation of serotonin.

**Figure 2 ijms-23-00010-f002:**
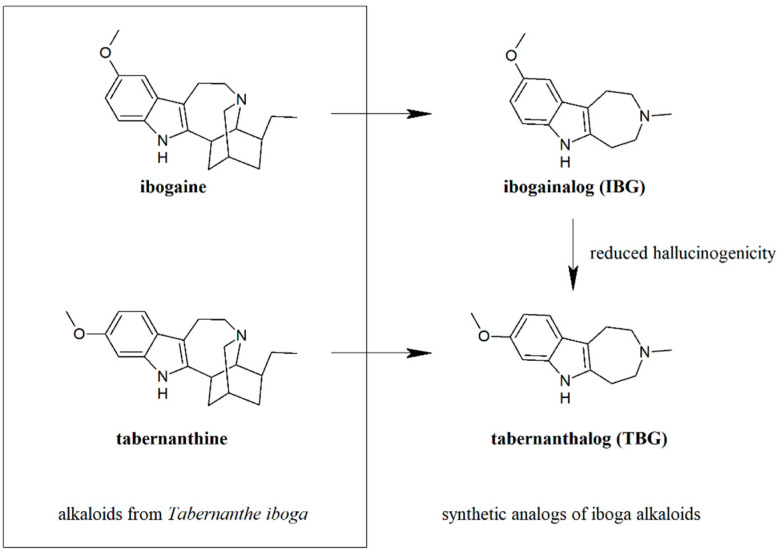
Synthetic 5-HT_2A_ agonists and their precursors of natural origin.

**Table 1 ijms-23-00010-t001:** Classification of 5-HT receptors according to the affected signaling pathways and pharmacological importance.

Group	Receptor Subtype	Pharmacological Importance
G_αi_-coupled serotonin receptors; usually decrease the activity of adenylyl cyclase and lower the intracellular cAMP concentration [32]	5-HT_1A_, 5-HT_1B_, 5-HT_1D_, 5-HT_1E_, 5-HT_1F_ 5-HT_5A_, 5-HT_5B_	Anxiety, novel antidepressants, antipsychotics, migraine Locomotion, sleep
G_αq/11_-coupled serotonin receptors; activate the phospholipase C, increase the concentration of inositol triphosphate, diacylglycerol, and intracellular calcium levels [32]	5-HT_2A_, 5-HT_2B_, 5-HT_2C_	Schizophrenia, depression, Tourette’s syndrome, appetite, addiction, anxiety, sleep, sexual behavior
G_αs_-coupled serotonin receptors; couple to adenylyl cyclase, elevate cAMP levels, may also result in increased calcium [32]	5-HT_4_, 5-HT_6_, 5-HT_7_	Respiration, sleep, thermoregulation, mood, learning, cognition, anxiety, appetite

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
