# Peer review of "Overcoming Depression with 5-HT2A Receptor Ligands"

_ijms, 2021, doi:10.3390/ijms23010010_

Round 1

Reviewer 1 Report

Review Zięba et al.

Overcoming depression with 5-HT2A receptor ligands

The most widespread monoamine of the CNS, serotonin, impacts a variety of functions. Serotonin is a crucial signal in the neurogenic niche microenvironment of the hippocampus and key player in antidepressant action that involves increased adult neurogenesis. Deregulation of the serotonin system leads to neurogenic decline, changes in appetite, mood disorders, and decreased metabolic rate. Serotonin receptors are targets for pharmacotherapy: in depression, antidepressants (SSRI) are thought to primarily inhibit serotonin reuptake into the presynaptic cell and to target 5-HT1A auto-receptors; 5-HT2 receptors can be targets following stroke and in obesity. In the current review, Zieba et al. summarized the role of serotonin in the CNS and focus especially on the 5-HT2A receptor subtype ligands as new target in neurodegenerative disorders. The authors summarize the receptors structure and function, and discuss its current and future role in mediating therapeutic benefits. The topic of the review is novel and interesting. Overall, the review needs some structural corrections, and a more precisely mention of the reviewed papers. Please find my general comments on the order, and specific indications to improve the relevance for a broader audience below.   

Overall comments include:

1. Please use abbreviations consistently throughout, i.e. introduce serotonin (5-hydroxytryptamine, 5-HT) at front and keep 5-HT throughout the document (start somewhere between line 46 and 56); 2. CNS is introduced several times; 3. Line 170 and further, three-dimensional, 3D; 4. AlphaFold; 5. I am not sure whether ‘depression’ can be characterized as ‘disease’ rather use ‘disorder’

Specific comments, Introduction:

Lines 46-143 needs a better organization; the authors start with depression i.e. before, and go back and forth on the discovery of serotonin, the receptors, synthesis, SSRIs. Line 56 to 71 the discovery seems not relevant for the review (discovery/name), shorten it, and also summarize in a few sentences the existence and differences of the two independent sources of serotonin (in the periphery vs brain) and add it’s function (81 to 101; and it is not ‘a’ tryptophan’, TPH is the main enzyme in serotonin synthesis); in the further paragraphs, keep it simple and summarize, also mention the importance of the hippocampus (main target of 5-HT fibers, see below line 128 ..) and the hypothalamus – since the authors write ‘appetite, sexual behavior’ etc. (line 92); line 82, 5-HT is not secreted and then stored, but released by vesicles; cut line 103 – 107, and add 5-HT receptors and its function (in more detail, i.e. introduced G-protein coupled fct here since it is relevant for the ligand studies); cut 115-127, its repetition; line 128-142 it’s about the receptors again.

I suggest to start/continue with the specific receptors (from line 153 on), especially 5-HT1A, 2A, and 2C (put lines 276 and further here, lines 430-40 also) – then more specifically why 2A has been chosen (lines 328 to 360) – importantly, the role in the hippocampus, i.e. on neurogenesis, is not stated here and only hinted later – REFs are needed and the role of 1A and 2A/C subtypes in antidepressant action; it should be more compact, and then continue with its structure (new headline) and role as hallucinogens, line 227, or psychedelic (line 361) – please put this together in simpler paragraphs. Maybe reconsider using questions as headlines and summarize the relevant literature per topic.

Line 154 take out ‘previously termed subtype’ and simply use 5-HT2A throughout;

Figure 1 is very simple, maybe one could add the different function of SSRIs, vs psychedelic vs hallucinogen? anyway, please put ‘postsynaptic 5-ht receptor’ outside the cell next to the receptors; and add ‘O’ to MAO in the legend

Author Response

Dear Reviewer,

We want to express our gratitude for taking your time to consider our manuscript. We are pleased that you shared your inspiring thoughts with us and helped us improve the submitted work. Moreover, we want to emphasize, we considered each of your suggestions and implemented several changes that improved the readability of the text. Please find our answers to your comments enclosed within this file.

Yours faithfully,
Agata Zięba and co-authors

Reviewer 2 Report

Please see some comments and corrections below.

Abstract: depression is not a complex disease but a multifactorial disorder. Novel compounds with antidepressant action, not necessarily antidepressants, as those need to be established in clinical trials / practice.

“Articles were critically assessed based on their titles and abstracts. The most accurate papers were chosen to be presented in the manuscript” were they even read or the information is only based on the titles and abstracts??

Why is “hallucinogens” a keyword?

“More than half of the people diagnosed with depression live in South-East Asia or Western Pacific Region.” Numbers? Reference? It is hard to believe because even though the highest populated countries are located in those regions, depression is more frequently assessed and diagnosed in so-called “developed countries”, and therefore in North America and Europe.

L64 – when you mention Cleveland, maybe would be better to indicate it is in the USA

L70 – depression and autism are not diseases.

Figure 1 – poor resolution of structure captions. Why write “M.A. stands for monoamine oxidase-an enzyme mainly responsible for the degradation of serotonin.” In the caption for figure 1? The abbreviation isn’t even used in the figure…

L92 / 99 – remove e.g. (do not use this abbreviation in current text, only between parenthesis)

Table 1 has a poor readout and correspondence between columns.

In line 119, when an association between subtypes and functions are expressed, please indicate which are responsible / associated with what.

L146 – is an “inverse agonist” the same as an antagonist?

L161 – Yes, located on chromosome 13, but which branch and position?

L178 – please provide the date of Jaiteh et al paper.

L233 – Many studies… please provide references…

L356 – what is the “low rate 72 seconds schedule”

L391 – “The quantity of papers focusing on this field is growing” – please provide numbers/metrics and a timescale (1y, 10y…), instead of a selection of reference

L441 – This is not a chapter, but a review. Or was this material published elsewhere? Or was this integrally copied?

L522 – how does all that translate to ADME? Did not understand exactly what that acronym is for.

L525 – again, there are no chapters. It is a review.

L546 – What is a QSAR model?

L544-557 – This is very random, with little context and seemingly unintelligible. There is no context to these databases, what are they about? What are these predictions? And models?

571-586 – Same thing going on here… LigandScout? TCMNP? MD simulation performances?

Author Response

(The authors gave the same response as above.)

Round 2

Reviewer 1 Report

Dear Authors,

I am happy to see my concerns addressed.

However, there is a minor, yet severe mistake in the 5th paragraph. Serotonin neurons are mainly located in the brain stem raphe nuclei, not the regions described by the authors! while fibers spread throughout the brain, and mostly release the neurotransmitter en passent into the limbic system. I have revised the paragraph, and added the appropriate Ref, and it could look like this:

Line 83 to 97:

Serotonin is an amine whose action is mainly linked with the central nervous system (CNS) effects. Serotonergic neurons are mainly located in the brain stem raphe nuclei (Gaspar P et al. Nat Rev Neurosci 2003). Fibers terminate throughout the CNS, and the transmitter is released by synaptic vesicles. Neuroimaging studies show highest density of serotonin in limbic structures – cingulate, entorhinal, insular, and temporopolar regions, along with the ventral and pallidal regions of the striatum and the medial orbitofrontal cortex [13]. High density of serotonin is also identified in hippocampus. It is worth highlighting that the mentioned structures resemble a “social brain”, comprising regions important for social cognition and decision making [14]. Within the CNS, the released serotonin binds to specific proteins to activate 5-HT receptors located on postsynaptic neurons or serotonin autoreceptors located on the presynaptic part of the neuronal membrane. Serotonin and its receptors are believed to regulate, e.g. sexual behavior [14], mood [15], cog- nition [16], appetite [17], anxiety [8], aggression [18], the release of other neurotransmitters, addiction development [19], and sleep-wake cycles [20].

Author Response

Dear Reviewer,

Thank you for performing an additional evaluation of our manuscript. We appreciate all given suggestions, and we believe that they significantly increased the readability of this work. Regarding the last call for modifications, we decided to reformulate lines 78-79:

“Serotonin is an amine whose action is mainly linked with the central nervous system (CNS) effects and its neurons are mainly located in the brain stem raphe nuclei [13]. Fibers terminate throughout the CNS, and the transmitter is released by synaptic vesicles. Neuroimaging studies confirm that the highest density of serotonin neurons can be identified in limbic structures – cingulate, entorhinal, insular, and temporopolar regions, along with the ventral and pallidal regions of the striatum and the medial orbitofrontal cortex [14]. High density of serotonin neurons is also identified in the hippocampus. It is worth highlighting that the mentioned structures resemble a “social brain”, comprising regions important for social cognition and decision making [15]. Within the CNS, this substance is released into the synaptic cleft and binds to specific proteins to activate 5-HT receptors located on postsynaptic neurons or serotonin autoreceptors located on the presynaptic part of the neuronal membrane. Serotonin and its receptors are believed to regulate e.g. sexual behavior [15], mood [16], cognition [17], appetite [18], anxiety [8], aggression [19], the release of other neurotransmitters, addiction development [20], and sleep-wake cycles [21]”.

Yours faithfully,
Agata Zięba and co-authors

Reviewer 2 Report

The authors have addressed the concerns raised in the previous submission and improved the overall quality of the manuscript, which is now suitable for publication.

Author Response

Dear Reviewer,

Thank you for taking the time to review our manuscript. We sincerely appreciate all valuable comments and suggestions, which helped us to improve the quality of this article. We are pleased that you consider our article suitable for publication.

Yours sincerely,
Agata Zięba and co-authors